# Spatial transformation of multi-omics data unlocks novel insights into cancer biology

Mateo Sokač[1,2,3], Asbjørn Kjær[1,2,3], Lars Dyrskjøt[1,2], Benjamin Haibe-Kains[4], Hugo JWL Aerts[5,6,7], Nicolai J Birkbak[1,2,3]*

[1]Department of Molecular Medicine, Aarhus University Hospital, Aarhus, Denmark; [2]Department of Clinical Medicine, Aarhus University, Aarhus, Denmark; [3]Bioinformatics Research Center, Aarhus University, Aarhus, Denmark; [4]Princess Margaret Cancer Centre, University Health Network, Temerty Faculty of Medicine, University of Toronto, Toronto, Canada; [5]Artificial Intelligence in Medicine (AIM) Program, Mass General Brigham, Harvard Medical School, Boston, United States; [6]Departments of Radiation Oncology and Radiology, Brigham and Women's Hospital, Dana-Farber Cancer Institute, Harvard Medical School, Boston, United States; [7]Radiology and Nuclear Medicine, CARIM & GROW, Maastricht University, Maastricht, Netherlands

*For correspondence:
nbirkbak@clin.au.dk

**Competing interest:** The authors declare that no competing interests exist.

**Abstract** The application of next-generation sequencing (NGS) has transformed cancer research. As costs have decreased, NGS has increasingly been applied to generate multiple layers of molecular data from the same samples, covering genomics, transcriptomics, and methylomics. Integrating these types of multi-omics data in a combined analysis is now becoming a common issue with no obvious solution, often handled on an ad hoc basis, with multi-omics data arriving in a tabular format and analyzed using computationally intensive statistical methods. These methods particularly ignore the spatial orientation of the genome and often apply stringent p-value corrections that likely result in the loss of true positive associations. Here, we present GENIUS (GEnome traNsformatIon and spatial representation of mUltiomicS data), a framework for integrating multi-omics data using deep learning models developed for advanced image analysis. The GENIUS framework is able to transform multi-omics data into images with genes displayed as spatially connected pixels and successfully extract relevant information with respect to the desired output. We demonstrate the utility of GENIUS by applying the framework to multi-omics datasets from the Cancer Genome Atlas. Our results are focused on predicting the development of metastatic cancer from primary tumors, and demonstrate how through model inference, we are able to extract the genes which are driving the model prediction and are likely associated with metastatic disease progression. We anticipate our framework to be a starting point and strong proof of concept for multi-omics data transformation and analysis without the need for statistical correction.

## eLife assessment

This **valuable** manuscript presents a new approach to transform multi-omics datasets into images and to exploit Deep Learning methods for image analysis of the transformed datasets. As an example, the method is applied to multi-omics datasets on different cancers. While the evidence in this specific case is **solid**, whether the method is working as advertised in other settings is not yet known.

## Introduction

The recent advent of next-generation sequencing (NGS) has revolutionized research and has been applied extensively to investigate complex biological questions. As the cost of sequencing continues to drop, it has become increasingly common to apply NGS technology to investigate complementary aspects of the biological processes on the same samples, particularly through analysis of DNA to resolve genomic architecture and single-nucleotide variants, RNA to investigate gene expression, and methylation to explore gene regulation and chromatin structure. Such multi-omics data provides opportunities to perform integrated analysis, which investigates multiple layers of biological data together. Over the years, this has resulted in the generation of an incredible amount of very rich data derived from the genome itself, either directly or indirectly. The genome is spatially organized, with genes positioned on chromosomes sequentially and accessed by biological processes in blocks based on chromatin organization (*Franke et al., 2016*). However, genome-derived NGS data is usually stored in and analyzed from a tabular format, where the naturally occurring spatial connectivity is lost. Furthermore, while genomic data is rich, the feature space is generally much larger than the number of samples. As the number of features to evaluate in statistical tests increases, the risk of chance associations increases as well. To correct for such multiple hypothesis testing, drastic adjustments of p-values are often applied which ultimately leads to the rejection of all but the most significant results, likely eliminating a large number of weaker but true associations. While this is a significant issue when analyzing a single type of data, the problem is exacerbated with performing multi-omics analysis where different types of data are combined, often in an ad hoc manner tailored to specific use cases. Importantly, a common theme in multi-omics analytical approaches is that observations are processed individually, thereby discarding potential spatial information that may originate from the organization of genes on individual chromosomes.

Using artificial intelligence methods may help overcome this problem. Over the past decade, the development of artificial intelligence methods, particularly within deep learning architectures, has thoroughly revolutionized several technical fields, such as computer vision, voice recognition, advertising, and finance. Within the medical field, the roll-out of AI-based technologies has been slower, hampered in part by considerable regulatory hurdles that have proven difficult for machine-learning applications where the systems may accurately classify patients or samples by some parameter, but the logical reason behind this is unclear (*Wiens et al., 2019*). Nevertheless, AI systems have proven successful in a multitude of medical studies, and in recent years some AI-powered tools have started to move past testing to deployment (*Benjamens et al., 2020*). A major benefit of deep neural networks is that they can capture nonlinear patterns in the data without necessitating correction for multiple hypothesis testing. Additionally, the use of convolutional layers within the networks has shown to improve performance by decreasing the impact of noise (*Jang et al., 2021*; *Du et al., 2022*). However, the problem with complex deep learning models is not the analysis itself but their interpretation (*Rudin, 2019*). Simpler models tend to have high interpretability; however, they are unable to capture complex nonlinear connections in data. This often leads to the utilization of 'black box' models at the cost of interpretability (*Elmarakeby et al., 2021*; *Wolfe et al., 2021*). 'Black box' models are popular in the artificial intelligence industry, especially in computer vision applications, where immense progress is being made in technologies such as self-driving cars and computer interpretation of images. However, in many of those applications, the interpretability of models is not as important as in medicine (*Yang et al., 2022*; *Petch et al., 2022*).

In medicine, the interpretability of models is crucial since there is a need for discovering new biomarkers as well as identifying underlying biological processes (*Picard et al., 2021*). In addition to advancements in artificial intelligence and NGS, a vast amount of research has been conducted to interpret highly complex machine-learning models; frameworks such as DeepLIFT (*Shrikumar et al., 2017*), Integrated Gradients (IG; *Ancona et al., 2017*; *Sundararajan et al., 2017*), and DeepExplain (*Shrikumar et al., 2017*; *Samek et al., 2019*; *Bach et al., 2015*) were developed in recent years with the purpose of debugging complicated machine-learning models (*Despraz et al., 2017*). These frameworks enable the usage of deep learning models for integrated multi-omics analysis through their ability to evaluate input attribution in models that are traditionally considered a 'black box'. In multi-omics analysis, this means that it is possible to combine the entirety of the data from multiple data sources into a high-dimensional data structure and process it with deep learning models without

losing interpretability. As output, an attribution score can be produced for every input, which may be interpreted as the relative importance of the feature in the model and used for further analysis.

Here, we present a framework for multi-omics analysis based on a convolutional deep learning network to find hidden, nonlinear patterns in spatially connected feature-rich multi-layered data. The spatial connection of the data is made by transforming the data into a multi-channel image in such a way that spatial connections between genes are captured and analyzed using convolutional layers. Using spatial connections between the data showed superior performance when compared to non-spatially data transformations. Furthermore, the trained model is combined with IG, which allows us to evaluate the relative contribution of individual features and thus decipher the underlying biology that drives the classification provided by the deep learning models. IG is a non-parametric approach that evaluates the trained model relative to input data and output label, resulting in attribution scores for each input with respect to the output label. In other words, IG represent the integral of gradients with respect to inputs along the path from a given baseline. By using IG, we provide an alternative solution to the problem posed by performing multiple independent statistical tests. Here, instead of performing multiple tests, a single analysis is performed by transforming multi-omics data into genome images, training a model, and inspecting it with IG. IG will output an attribution score for every gene included in the genome image. These can be ranked in order to retrieve a subset of the most associated genes relative to the output variable. We named the framework GENIUS (GEnome traNsformatIon and spatial representation of mUltiomicS data), and the methodology may be split into two parts, classification and interpretation. First, the key feature of GENIUS is that for classification, multi-omics data is transformed into multi-channel images where each gene is presented as a pixel in an image that covers the whole genome (*Figure 1A, B*). We then incorporate multiple types of omics data, such as mutation, expression, methylation, and copy number data, into the image as distinct layers. These layers are then used as input into the deep learning model for training against a binary or continuous outcome variable. Next, for interpretation, an attribution score is assigned to each feature using IG, allowing the user to extract information about which feature or features may drive a specific prediction based on deep learning analysis of input from multiple-omics data sources. In this work, we describe the development of the GENIUS framework and demonstrate its utility in predicting the development of metastatic cancer, patient age, chromosomal instability, cancer type, and as proof of concept, loss of TP53.

All predictions are based on multi-omics input through the GENIUS framework. Users may train their own or publicly sourced multi-omics data against a specified endpoint tailored to the user's choice. The GENIUS framework thus overcomes the issue of multiple hypothesis testing and may provide new insights into the biology behind classification by deep learning models. The GENIUS framework is made available as a GitHub repository and may be used without restrictions to develop stratification models and inform about genome biology using multi-omics input.

## Methods
### GENIUS model architecture and hyperparameters

We designed a four-part convolutional neural network with the purpose of extracting the features from multi-dimensional data while minimizing the impact of noise in the data (*Figure 1C*). The network was implemented using the PyTorch framework. The structure of the network is similar to an autoencoder architecture; however, the reconstruction of the genome image is not penalized. The motivation behind the implemented network structure is to use an encoder in order to learn how to compact genomic information into a small vector, L, forcing the network to extract relevant information from up to five data sources. The next module reconstructs the image from vector L, learns which features are important, reorganizes them optimally, and removes noise. The final module of the network uses a series of convolutions and max-pooling layers in order to extract information from the reconstructed image and, finally, predicts the outcome variable using a fully connected dense network.

The first part of the network is called the encoder, as its purpose is to encode the entire image to a vector of size 128, representing the latent representation of the input data, 'L'. Next, the original image is reconstructed from L into its original size using a decoder module in the network. In this step, since we are not using the reconstruction loss, the network reconstructs the image of a genome which is optimal for information extraction. This is followed by the extractor module containing convolution

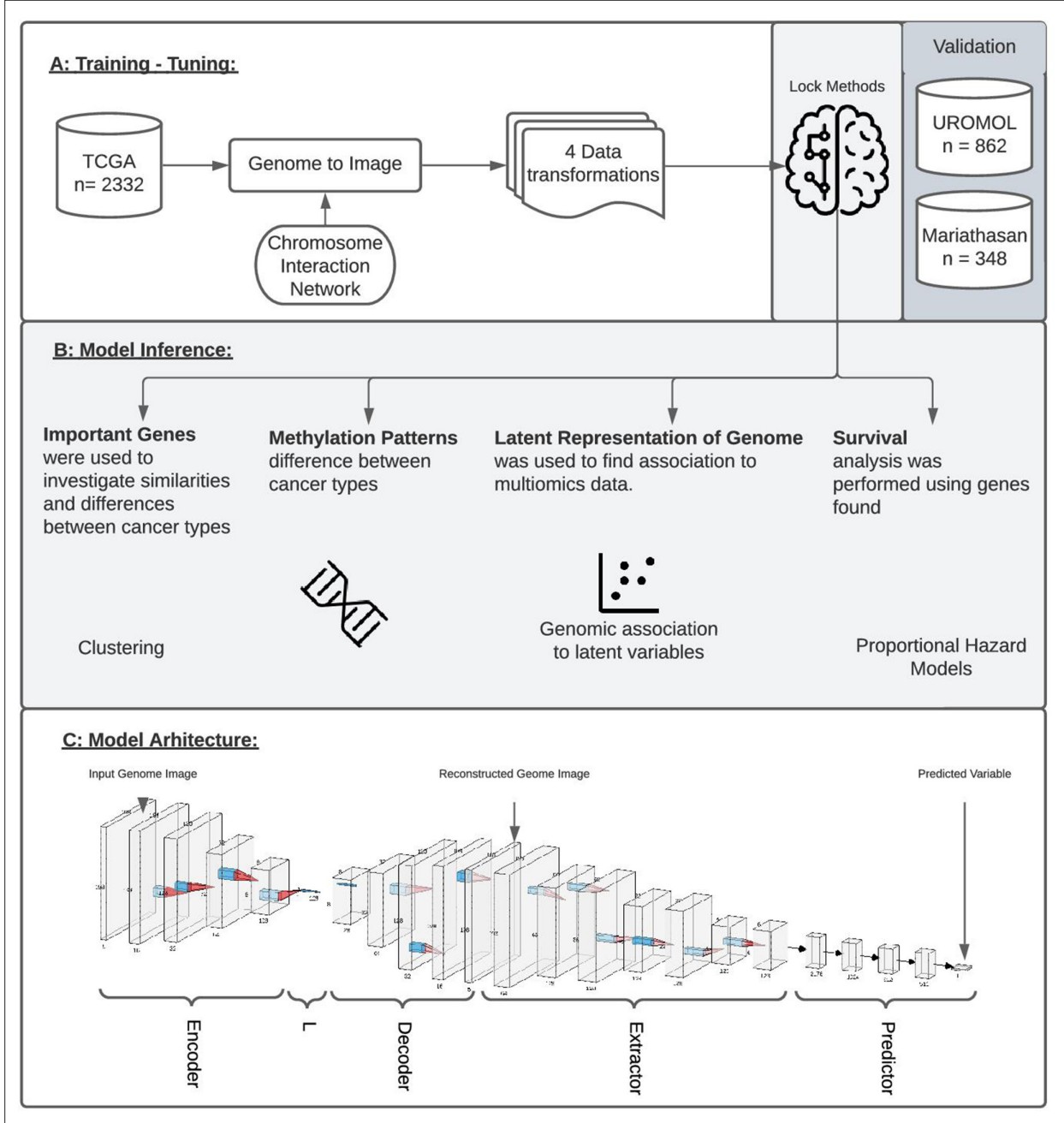

**Figure 1.** Study overview. (**A**) The study utilized 2332 tumor samples representing six cancer types (bladder, uterine, stomach, ovarian, kidney, and colon) and transformed multi-omics data into images based on chromosome interaction networks. After the model was trained, we validated found genes with two independent cohorts representing early-stage bladder carcinoma (BLCA; UROMOL) and late-stage BLCA (Mariathasan). (**B**) The validation included looking at the most important genes driving metastatic disease, similar/different methylation patterns between cancer types, latent representation of genome data and looking at survival data. (**C**) The model architecture where the first part of the network encodes genome data into latent vector, L, followed by decoding where image is reconstructed. Next layers aim to extract information from the reconstructed image, concat it with L and make a final prediction.

The online version of this article includes the following figure supplement(s) for figure 1:

**Figure supplement 1.** Figure representing an example of genome image construction.

**Figure supplement 2.** Overall representation of a process where chromosomal instability information was included in genome image construction.

**Figure supplement 3.** Figure showing the percent of samples classified as metastatic when using the stage as a variable.

and max-pooling layers aiming to extract relevant information from the reconstructed image. The final part of the network flattens the learned features obtained from previous layers, concatenates them with the L vector, and forwards it to a fully connected dense feed-forward network where the final prediction is made (*Figure 1C*; *LeNail, 2019*). During training, the last module of this model was adopted to predict qualitative as well as quantitative types of data.

All models were trained with Adagrad optimizer with the following hyperparameters: starting learning rate = 9.9e−05 (including learning rate scheduler and early stopping), learning rate decay and weight decay = 1e−6, batch size = 256, except for memory-intensive chromosome images where the batch size of 240 was used. Adding chromosome interaction information to the data transformation showed improvement during training; next question was whether we should penalize the reconstruction of genome image during the training process. After multiple training scenarios and hyperparameter exploration, we concluded that by forcing the network to reconstruct genome images in the process of learning, we are limiting network performance. Instead, we used the appropriate loss function for prediction and allowed the network to reconstruct genome images that are optimal for making predictions.

## Evaluating input image design

To evaluate the performance of GENIUS with an image-based transformation of input omics data, we tested four different image layouts of the genome. For each layout, we created a set of images where each sample is represented by one multi-channel image and each channel represented a specific type of omics data (gene expression, methylation, mutation, deletion, and amplification) (*Figure 1A*, *Figure 1—figure supplement 1A-B*). Each data type was encoded for each gene as a continuous value, where each gene was defined by a single pixel in each layer. We then tested the performance of the deep neural network on four different image layouts. First, we assembled the genome as a square image, measuring 198x198 pixels in total. Here, all genes were placed on the image sequentially according to their chromosomal locations, and individual chromosomes were organized by how close they were oriented in 3D space (*Sarnataro et al., 2017*). Second, we tested an image organized by 24 × 3760 pixels, with 3760 pixels representing the most gene-rich chromosome, and each chromosome placed below the other on the image following the same order as in 198 × 198 images. Chromosomes containing fewer than 3760 genes had black pixels added to the end to create a rectangular image. Third, we tested a random 2D location, with each gene placed as a random pixel in a 198 × 198 pixel square image. Lastly, we tested an image of a single vector with all genes placed in a randomly ordered sequence. Data transformation we performed and tested:

1. Square image (198 × 198 pixels), each gene represented by one pixel ordered by chromosome position. Chromosomes are ordered by interaction coefficient based on Hi-C sequencing (*Sarnataro et al., 2017*).
2. Square image (198 × 198 pixels), each gene is represented by one pixel located on the image in random order; thus, the 2D location carries no information.
3. Rectangular image (24 × 3760 pixels), each gene represented by one pixel ordered by chromosome position. Chromosomes are ordered by interaction coefficient based on Hi-C sequencing (*Sarnataro et al., 2017*).
4. A flat, one-dimensional vector containing all features from the five data sources in random order.

By using different image layouts, we wanted to investigate the spatial dependency of observations. Images were created by making a matrix for each source of data where each cell was represented by a single gene (*Figure 1A*, *Figure 1—figure supplements 1 and 2*). The genes in 198 × 198 and 24 × 3760 images were ordered by position as well as by chromosome interaction coefficients resulting in the following order of chromosomes: 4, X, 7, 2, 5, 6, 13, 3, 8, 9, 18, 12, 1, 10, 11, 14, 22, 19, 17, 20, 16, 15, 21. Finally, newly created observations for each data source were merged as a multi-channel image where each channel represents a single source of data (*Figure 1A*, *Figure 1—figure supplement 1*).

## Samples and training data

We obtained gene expression, exome mutation, methylation, and copy number data from six cancer types from the Cancer Genome Atlas (TCGA). These were picked to filter out cancer types with less than 400 samples. Next, cancer types with an extremely high or extremely low proportion of metastatic samples (0.85 < Proportion > 0.15) were removed, resulting in ovarian serous cystadenocarcinoma

(OV), colon adenocarcinoma (COAD), uterine corpus endometrial carcinoma (UCEC), kidney renal clear cell carcinoma (KIRC), urothelial bladder carcinoma (BLCA), and stomach adenocarcinoma (STAD) (*Figure 1A*, *Figure 1—figure supplement 3*). RNAseq was obtained from the University of California Santa Cruz (UCSC) Toil pipeline (*Vivian et al., 2017*) and summarized to transcript per million (TPM) on the gene level. SNP6 copy number data were segmented using ASCAT v2.4 (*Adzhubei et al., 2010*; *Raine et al., 2016*) and converted into a ploidy and purity normalized log *R* value by dividing the total copy number with ploidy and taking the log2 value. The weighted genome integrity index (wGII) (*Burrell et al., 2013*) was calculated on the available segmented copy number data, as previously described. Mutation calls were annotated using Polyphen2 to assess the mutation's impact on the protein structure. Methylation was summarized by the mean methylation score for each gene.

## Validation cohorts acquisition and processing

Two independent cohorts of bladder cancer patients were used for validation. The UROMOL cohort (*Zuiverloon et al., 2013*; *Lindskrog et al., 2021*) contains molecular data from 535 tumors from patients with early-stage bladder cancer (Ta and T1) and was included to evaluate the progression to muscle-invasive bladder cancer. The Mariathasan cohort (*Mariathasan et al., 2018*) contains molecular data from 348 tumors from patients with advanced or metastatic bladder cancer (stages III and IV), treated with checkpoint immunotherapy. This cohort was included to evaluate the ability of the GENIUS framework to predict the likelihood of developing metastatic disease.

For both cohorts, RNAseq data was aligned against hg38 using STAR (*Dobin et al., 2013*) version 2.7.2 and processed to generate count and TPM expression values with Kallisto (*Ayers et al., 2017*) version 0.46.2. Whole exome sequence data was processed using GATK (*Van der Auwera, 2020*) version 4.1.5 and ASCAT version 2.4.2 to obtain mutation and allele-specific copy number, purity, and ploidy estimates.

## Data transformation

All mutations were ranked by PolyPhen scores, ranging between 0 and 1. Log *R* segmented copy number data was analyzed as deletion and amplification separately. Copy number deletion was defined as log *R* scores <log2 of 0.5/2, copy number amplification was defined as log *R* scores >log2 of 5/2. All data types were defined on the gene level. For copy number alterations, we defined genes as amplified if the entirety of the gene was found within the amplified DNA segment. Genes were defined as deleted if they were partially or wholly within the deleted DNA segment (*Figure 1A*, *Figure 1—figure supplement 2A, B*). Finally, to enable data integration and for more stable training of machine-learning models, we generated mathematically equivalent values for each data source ranging from 0 to 1 through a simple linear transformation (min–max scaling). This enabled comparisons between individual data types, and was performed on each data source.

## Training scenarios

We used the GENIUS framework to make six models predicting the following conditions:

1. Metastatic cancer (binary classification), defined as stage IV versus stages I–III.
2. TP53 mutation (binary classification), where the TP53 mutation was removed from the input data and used only as a binary outcome label.
3. The tissue of origin (multi-class classification).
4. Age (continuous variable).
5. wGII (*Burrell et al., 2013*), a chromosomal instability marker (continuous variable).
6. Randomized tissue of origin (multi-class variable). By randomizing the tissue of origin labels, a negative control was created. The purpose of this negative control was to confirm the model would fail to predict a pattern when none existed.

In order to adapt the network for predicting different variables, we simply changed the output layer and loss function for training. Binary classifications and the multi-class classification used softmax as the output layer and the cross entropy loss function. When predicting continuous values, the model used the output from the activation function with the mean squared error loss function. When predicting multi-class labels, the performance measure was defined by the *F*1 score, a standard measure for multi-class classification that combines the sensitivity and specificity scores and is defined as the harmonic mean of its precision and recall. To evaluate model performance against the binary

outcome, ROC analysis was performed, and the area under the curve (AUC) was used as the performance metric.

### Latent representation of genome

The purpose of latent vectors is to capture the most significant information from the entire genome data and compress it into a vector of size 128. This vector was later appended into a feed-forward network when making the final prediction. This way, the model had access to extracted information before and after image reconstruction. After successful model training, we extracted the latent representations of each genome and performed the Uniform Manifold Approximation and Projection (UMAP) of the data for the purpose of visual inspection of a model (*Figure 2A, B*). The UMAP projected latent representations into two dimensions which could then be visualized. In order to avoid modeling noise, this step was used to inspect if the model is distinguishing between variables of interest. We observed that all training scenarios successfully utilized genome images to make predictions with the exception of Age, where no pattern was found from the genomic data, and randomized cancer type, which served as negative control where no pattern was expected (*Figure 2B*). Information in latent vectors extracted from Age-Model and randomized cancer type showed no obvious patterns, which is likely the cause of poor performance (*Figure 2—figure supplement 2A-B*).

### Identifying genes relevant to the tested outcome

Once the model was trained on the data, the appropriate loss function and output layer, including the model weights, were stored in a .pb file. The model and final weights were analyzed using the IG method implemented by Capture (*Sundararajan et al., 2017*). IG is an attribution method that assigns an 'attribution score' to each feature of the input data based on predictions the model makes. The attribution score is calculated based on the gradient associated with each feature of each image channel with respect to the output of the model. This information indicates to the neural network the extent of weight decrease or increases needed for certain features during the backpropagation process. Next, the created attribution images are used to extract information for each image channel and for every pixel. Since the pixels represent individual genes, this information can be reformatted and filtered to show the most important genes from every data source included in the analysis. All attribution scores were scaled using a min–max scaler for every cancer type to address biological differences between cancer types.

### Code availability

All code is available on the public GitHub repository (https://github.com/mxs3203/GENIUS; copy archived at *Sokač, 2023*), where the framework is easily available for analysis of private or public data. The framework provides tools to transform gene-oriented data into an image, train a model using existing model architecture and infer the most informative genes from the model using IG. The GitHub repository contains example data and instructions on how to use the GENIUS framework.

### Computational requirements

In order to train the model, we used the following hardware configuration: Nvidia RTX3090 GPU, AMD Ryzen 9 5950X16 core CPU, and 32 Gb of RAM memory. In our study, we used a batch size of 256, which occupied around 60% of GPU memory. Training of the model was dependent on the output variable. For metastatic disease prediction, we trained the model for approximately 4 hr. This could be changed since we used early stopping in order to prevent overfitting. By reducing the batch size to smaller numbers, the technical requirements are reduced making it possible to run GENIUS on most modern laptops.

## Results

### Building genome images to utilize spatial connections in genomic data

We endeavored to present genomic data as an image with genes represented as individual pixels to be processed by our deep learning architecture. To evaluate the relevance of the spatial orientation of the genes relative to model performance, we tested four different image layouts (Methods, *Figure 2A*): (1) Square image (198 × 198 pixels), each gene represented by one pixel ordered by

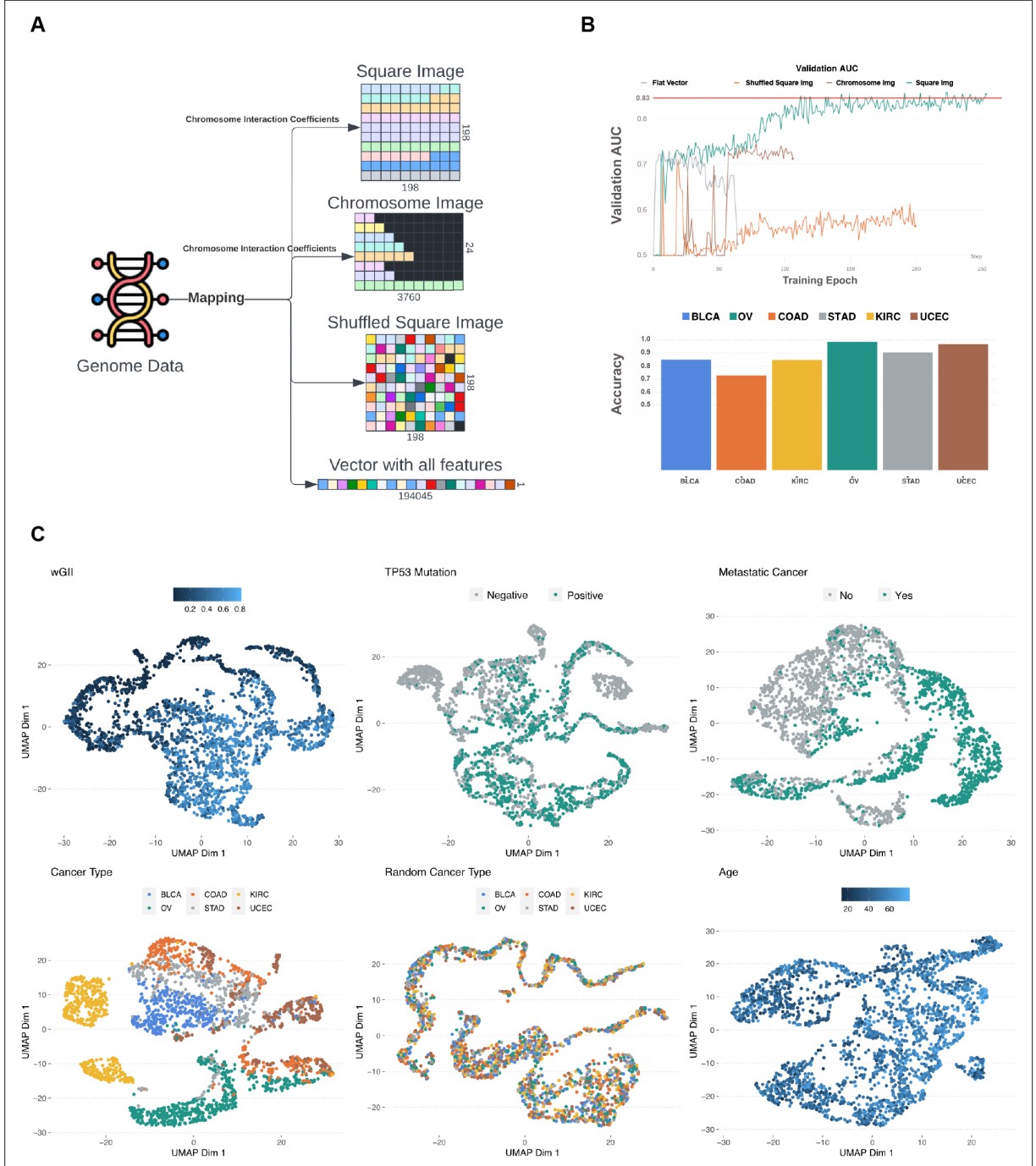

**Figure 2.** Data transformation overview. (**A**) The multi-omics genome data was transformed into four image types: square image organized by chromosome interaction network, chromosome image organized by chromosome interaction network, randomly organized image, and flat vector containing all multi-omics data. (**B**) The *x*-axis represents epochs and the *y*-axis represents area under the curve (AUC) score of fixed 25% data we used for accuracy assessment within the TCGA cohort. All four image types were used in training for metastatic disease prediction and the square image organized by chromosome interaction network resulted in best model performance (green color). The red line shows where the model resulted in the best loss. All curves stopped when the loss started increasing, indicating overfitting. The bar plot shows the proportion of correctly predicted (metastatic disease) in every cancer type included in the study. (**C**) Two-dimensional representation of vector L using Uniform Manifold Approximation and Projection (UMAP) for each predicted variable. Colors indicate the output variable which was used in the specific run.

The online version of this article includes the following figure supplement(s) for figure 2:

**Figure supplement 1.** Figure showing the process of training four image transformations in classification scenarios.

*Figure 2 continued on next page*

*Figure 2 continued*

**Figure supplement 2.** Figure showing the process of training four image transformations in regression scenarios.

**Figure supplement 3.** Figure showing the process of training four image transformations in classification scenarios (negative control).

chromosome position. Chromosomes are ordered by interaction coefficient based on Hi-C sequencing (*Yang et al., 2022*). (2) Square image (198 × 198 pixels), each gene is represented by one pixel located on the image in random order; thus, the 2D location carries no information. (3) Chromosome image (24 × 3760 pixels), each gene represented by one pixel ordered by chromosome position. Chromosomes are ordered by interaction coefficient based on Hi-C sequencing (*Yang et al., 2022*). (4) A flat, one-dimensional vector containing all features from the five data sources in random order. To evaluate the image layout, we used each type of layout to train against six biological states: (1) metastatic disease (stage IV vs. I–III), (2) cancer type, (3) burden of copy number alterations (defined by the wGII), (4) patient age, (5) TP53 status (where the TP53 pixel was set to '0' for all samples), and (6) randomized tissue type (negative control) (*Figure 2A*, *Figure 2—figure supplement 1A-E*). Every model output variable was trained until we observed no change in loss function or until validation loss values started increasing, indicating overfitting, which we handled by implementing early stopping.

While predicting metastatic disease, we observed that the Square Image data transformation outperformed all other data transformations, reaching a validation AUC of 0.87. The chromosome image and shuffled squared image performed similarly with AUC of 0.72 and 0.70, respectively (*Figure 2B*). Interestingly, the flat vector of features scored validation AUC around 0.84; however, the loss function started increasing as training epochs increased, indicating that the model was overfitted (*Figure 2B*, *Figure 2—figure supplement 1*). In the second scenario, we tested multi-class prediction using six cancer types in our dataset. Square Image outperformed other image layouts, reaching an $F1$ score of 0.81. Chromosome Image followed with an $F1$ score of 0.74, and the flat vector of features performed similarly to the random square image, reaching $F1$ scores of 0.66 and 0.71, respectively (*Figure 2A*, *Figure 2—figure supplement 1*). In order to address the framework's capabilities for predicting numeric output variables, we used wGII and patient age. Predicting wGII showed that the flat vector of features reached the least favorable Root Mean Squared Error (RMSE) score of 0.22, where chromosome image, shuffled square image, and square image reached similar RMSE scores of 0.16, 0.15, and 0.14, respectively (*Figure 2A*, *Figure 2—figure supplement 2*).

These results suggest that data layout does not play a major role when predicting wGII as the number of events in the genome would be predictive regardless of location. The age prediction model using square image data transformation outperformed other data transformations and obtained a validation RMSE of 0.19. The shuffled image performed the worst, reaching an RMSE of 0.49, while the chromosome-organized image and flat vector of features scored similar RMSE values of 0.38 and 0.31, respectively (*Figure 2A*, *Figure 2—figure supplement 2*). Additionally, the flat vector was inconsistent during training, but it did outperform the chromosome image. In the fifth scenario, we predicted the TP53 mutation status but removed the TP53 mutation itself from the data. Square Image performed the best, reaching a validation AUC of 0.83, whereas no major difference could be observed between a flat vector of features and Chromosome Image, reaching a validation AUC of 0.75 (*Figure 2A*, *Figure 2—figure supplement 1*). Finally, we tested the framework by predicting randomized cancer types as the negative control. All data transformations had similar and poor results (*Figure 2A*, *Figure 2—figure supplement 3*). For each output variable, we trained four different models utilizing the four data transformations. In all cases, the square image (198 × 198, ordered by chromosomes) outperformed the other transformations and was chosen as the layout for the final GENIUS framework, which was used for all subsequent analyses.

## Latent representation of genome captures relevant biology

The model architecture contains an encoder and decoder connected by a latent vector of size 128 (L), which provides the opportunity to inspect model performance (*Figure 1C*). The L vector is considered the latent representation of the genome data because it extracts and captures the most relevant data with respect to the output variable. This implies that an optimally trained model would show a perfect latent representation of the genome when overlaid with the output variable. Furthermore, this vector was later appended into a feed-forward network when making the final prediction. This way, the model had access to extracted information before and after image reconstruction. In order

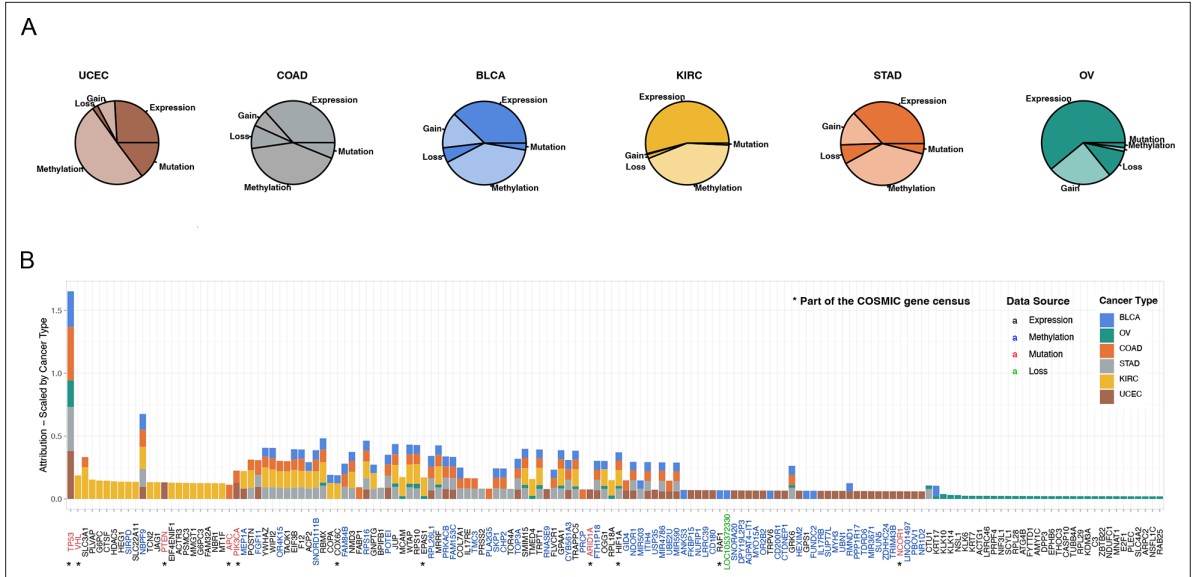

**Figure 3.** The most important events in metastatic disease development. (**A**) Pieplot showing the relative importance of each data source when predicting metastatic disease for each cancer type included in the study. (**B**) Top 50 genes for every cancer type scale by cancer type. The star symbol below the gene names indicates that the gene is part of COSMIC gene consensus. The color of the gene name indicates the data source and color of the bar indicates the cancer type.

The online version of this article includes the following figure supplement(s) for figure 3:

**Figure supplement 1.** The figure shows the relative attribution of each variable we predicted using the GENIUS framework.

**Figure supplement 2.** The figure shows the relative attribution of each variable we predicted using the GENIUS framework (*x*-axis) but split by cancer type.

**Figure supplement 3.** The figure shows patterns of methylation across chromosomes 1–22.

to visually inspect patterns captured by the model, we extracted the latent representations of each genome and performed the UMAP of the data to project it into two dimensions. We observed that all training scenarios successfully utilized genome images to make predictions that clustered into distinct groups, with the exception of Age. As expected, randomized cancer type, which served as negative control, also performed poorly (*Figure 2C*). Information in latent vectors extracted from Age-Model and randomized cancer type-model showed no obvious patterns, which is likely the cause of poor performance.

## GENIUS classification identifies tumors likely to become metastatic

To explore the utility of the GENIUS framework to classify tumors from multi-omics data and to interpret the biological drivers behind the classification, we further investigated the GENIUS model trained against metastatic disease using the TCGA datasets (*Figure 2B*). This analysis included primary tumors from six cancer types, a total of 2307 tumors, with 53% progressing to metastatic disease BLCA (277 metastatic/133 not-metastatic), OV (535 metastatic/47 not-metastatic), COAD (196 metastatic/254 not-metastatic), STAD (230 metastatic/189 not-metastatic), KIRC (208 metastatic/326 not-metastatic), and UCEC (117 metastatic/394 not-metastatic). The omics data types included somatic mutations, gene expression, methylation, copy number gain, and copy number loss. Using holdout type cross-validation, where we split the data into training (75%) and validation (25%), we observed a generally high performance of GENIUS, with a validation AUC of 0.83 for predicting metastatic disease (*Figure 2B*). The GENIUS framework allows us to explore the attribution of individual data layers to the final prediction. Across the cohort, gene expression and methylation data were generally the most informative data layers when it comes to classifying metastatic disease (*Figure 3A*). We noted that expression and methylation overall ranked the highest in terms of mean scaled attribution, with the exception of OV, which showed enrichment in methylation followed by copy number gain and loss. The same analysis was performed for cancer type, wGII, patient age, TP53 status, and randomized tissue type (*Figure 3A*, *Figure 3—figure supplements 1 and 2*).

## Interpreting the GENIUS model classifying metastatic cancer biology

Analyzing raw attribution scores we concluded the most informative data type overall regarding the development of metastatic disease was methylation (*Figure 3A*). To identify the individual genes driving the prediction, we pulled the 100 genes with the highest methylation attribution according to the GENIUS classification. We observed that many methylated regions overlapped between the six cancer types. These regions included methylation on specific regions of chromosomes 1, 6, 11, 17, and 19 (*Figure 3A*, *Figure 3—figure supplement 3*). Additionally, OV showed a unique methylation pattern spanning most of chromosome 7, while KIRC, COAD, and BLCA displayed regions of overlapping methylation on chromosome 22. We also noticed that mutation data often had a single mutation with a large attribution score while expression and methylation showed multiple genes with high attribution scores. To determine the genes that overall across the multi-omics data analysis contributed the most to the GENIUS classification of metastatic disease, we normalized gene attribution by cancer type and compared the top 50 genes for each cancer type (total of 152 genes, *Figure 3B*, *Supplementary file 1*). Unsurprisingly, we observed that TP53 mutations held the highest attribution score, followed by mutations to VHL. Both of these genes are well-established drivers of cancer and were previously reported as enriched in metastatic cancer (*Pandey et al., 2021*; *Christensen et al., 2022*), likely representing a more aggressive disease. However, of the 152 top genes, we noted only 11 genes previously reported as either oncogenes or tumor suppressor genes in the COSMIC cancer gene census (*Figure 3B*, indicated with a star), leaving 141/152 as potentially novel cancer genes. The highest scoring gene not previously associated with cancer was SLC3A1, the expression of which was found to be strongly associated with metastatic disease in clear cell renal cancer. SLC3A1 gene is a protein-coding gene associated with the transportation of amino acids in the renal tubule and intestinal tract, and aberrations in this gene have been associated with cystinuria, a metabolic disorder of the kidneys, bladder, and ureter (*Jiang et al., 2017*; *Woodard et al., 2019*). Furthermore, we identified PLVAP, often involved in MAPK cascades as well as in cellular regulatory pathways and the tumor necrosis factor-mediated signaling pathway. In BLCA, one of our most significant findings was increased expression of KRT17, a gene associated with a cytoskeletal signaling pathway, glucocorticoid receptor regulatory network, and MHC class II receptor activity (*Wu et al., 2021*; *Li et al., 2021*).

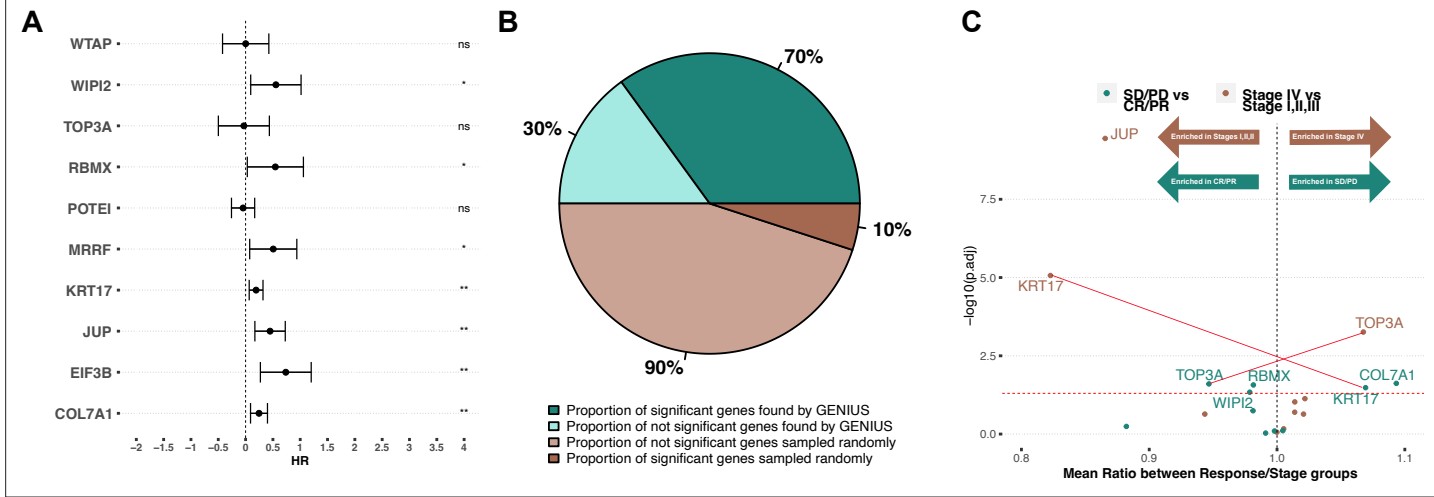

**Figure 4.** Validation on late-stage immunotherapy-treated bladder cancer (Mariathasan). (**A**) Forest plot showing top 10 expressed/methylated genes in multivariate cox proportional hazard model. X-axis indicates Hazard Rate (HR). Stars indicates significance (* P < 0.05, ** P < 0.01, *** P < 0.001), "ns" indicates not significant. (**B**) Comparison of median percent of randomly selected genes versus genes picked by GENIUS in cox proportional hazard model. (**C**) Volcano plot showing top 10 expressed/methylated genes and their enrichment in two comparisons; stages I, II and III versus stage IV and immunotherapy response (CR: complete response, PR: partial response) versus no response (SD: stable disease, PD: progressive disease). Two genes show association in opposite directions, indicated by red lines (KRT17, associated with low stage and poor immunotherapy response, and TOP3A, associated with high stage and improved immunotherapy response).

The online version of this article includes the following figure supplement(s) for figure 4:

**Figure supplement 1.** Histogram of correlation coefficients between each gene found in BLCA TCGA gene expression data and methylation data.

**Figure supplement 2.** Histogram showing distribution of number of randomly picked significant genes in Mariathasan dataset.

KRT17 has previously been reported as a potential cancer gene, but with an uncertain role (*Zhang et al., 2022*). Across cancer types, TOP3A was found to be commonly methylated in BLCA, COAD, STAD, and UCEC. TOP3A is associated with homology-directed repair and methylation may lead to increased chromosomal instability, a hallmark of cancer (*Hanahan and Weinberg, 2011*). The top 10 most important events driving the prediction of every output variable included in the study are summarized in *Supplementary files 2 and 3*.

## Validation of bladder cancer metastasis-associated genes in an independent cohort of advanced and metastatic bladder cancer

To investigate if the genes with the highest attribution score in the TCGA bladder cancer analysis were indeed associated with metastatic bladder cancer, we utilized an immunotherapy-treated predominantly late-stage (mainly stage III and IV) bladder cancer cohort with gene expression data available for 348 tumors (*Mariathasan et al., 2018*). For this analysis, we considered only the methylation and gene-expression-associated genes from the TCGA analysis. For methylation, we restricted the analysis to genes showing a significantly negative correlation between gene expression and gene-specific methylation levels (*Figure 4*, *Figure 4—figure supplement 1*). We then combined the methylation and gene-expression-based attribution scores and took the top 10 genes: RBMX, COL7A1, KRT17, JUP, WIPI2, TOP3A, EIF3B, WTAP, POTEI, and MRRF. Next, we implemented 10 multivariate Cox proportional hazard models (one for each gene), including available clinical parameters such as tumor stage, gender, neoantigen burden and baseline performance status (*Supplementary file 4*). This showed that in multivariate analysis, 7/10 genes had a significant association with outcome (*Figure 4A*). To evaluate the results of this analysis, we compared it to an identical model run 1000 times, but where the 10 genes were randomly picked. In 1000 runs, not one returned at least 7 significant genes (p < 0.001) (*Figure 4A*, *Figure 4—figure supplement 2*). The median percentage of significant genes for each run is reported in *Figure 4B*. Next, we performed two independent analyses, comparing the expression values of the top 10 genes between either (1) tumors defined as stage IV versus stages I and III, and (2) patients that responded to immunotherapy (CR and PR) versus patients that did not respond to immunotherapy (stable disease [SD] and progressive disease [PD]). Following correction for multiple hypothesis testing, we observed that TOP3A showed significantly increased expression in stage IV tumors, while JUP and KRT17 were significantly increased in stage I–III tumors (*Figure 4C*, brown dots). When comparing gene expression to response to immunotherapy, TOP3A, RBMX, and WIPI2 were significantly more expressed in complete response (CR)/partial response (PR) while KRT17 and COL7A1 were significantly more expressed in SD/PD. Interestingly, we observed increased expression of TOP3A in stage IV tumors, suggesting a role in metastatic disease, yet we also observed that the same gene was more expressed in tumors that responded to immunotherapy. This suggests that TOP3A is associated with the development of metastatic disease, but its expression may result in the development of a bladder cancer phenotype that is more sensitive to checkpoint immunotherapy.

## Validation of metastasis-associated genes in an independent cohort of early-stage bladder cancer

To investigate if the metastasis-associated genes found through the GENIUS framework also plays a role in the development of aggressive features in early-stage bladder cancer, we acquired the UROMOL dataset (*Lindskrog et al., 2021*), which includes gene expression data from 535 low-stage tumors. We again investigated the top 10 methylated or expressed genes found in the TCGA analysis of BLCA, using the gene expression data from UROMOL. First, we performed Cox proportional hazard analysis with progression-free survival (PFS) using the top 10 genes found by the GENIUS framework, again creating 10 individual models containing the selected genes and available clinical factors such as age, tumor stage, and sex. This showed that in multivariate analysis, 5/10 genes had a significant association with outcome (*Figure 5A*). The results were compared with cox proportional hazard models utilizing random sets of 10 genes, repeated 1000 times. Of these, 216 runs showed at least five significant genes (p = 0.216) (*Figure 5A*, *Figure 5—figure supplement 1*), indicating that in early-stage bladder cancer, the genes found by GENIUS to be associated with cancer metastasis were not uniquely relevant for disease progression. However, when we computed the median percentage of significant genes and compared it to the top 10 genes picked by the GENIUS framework, by random

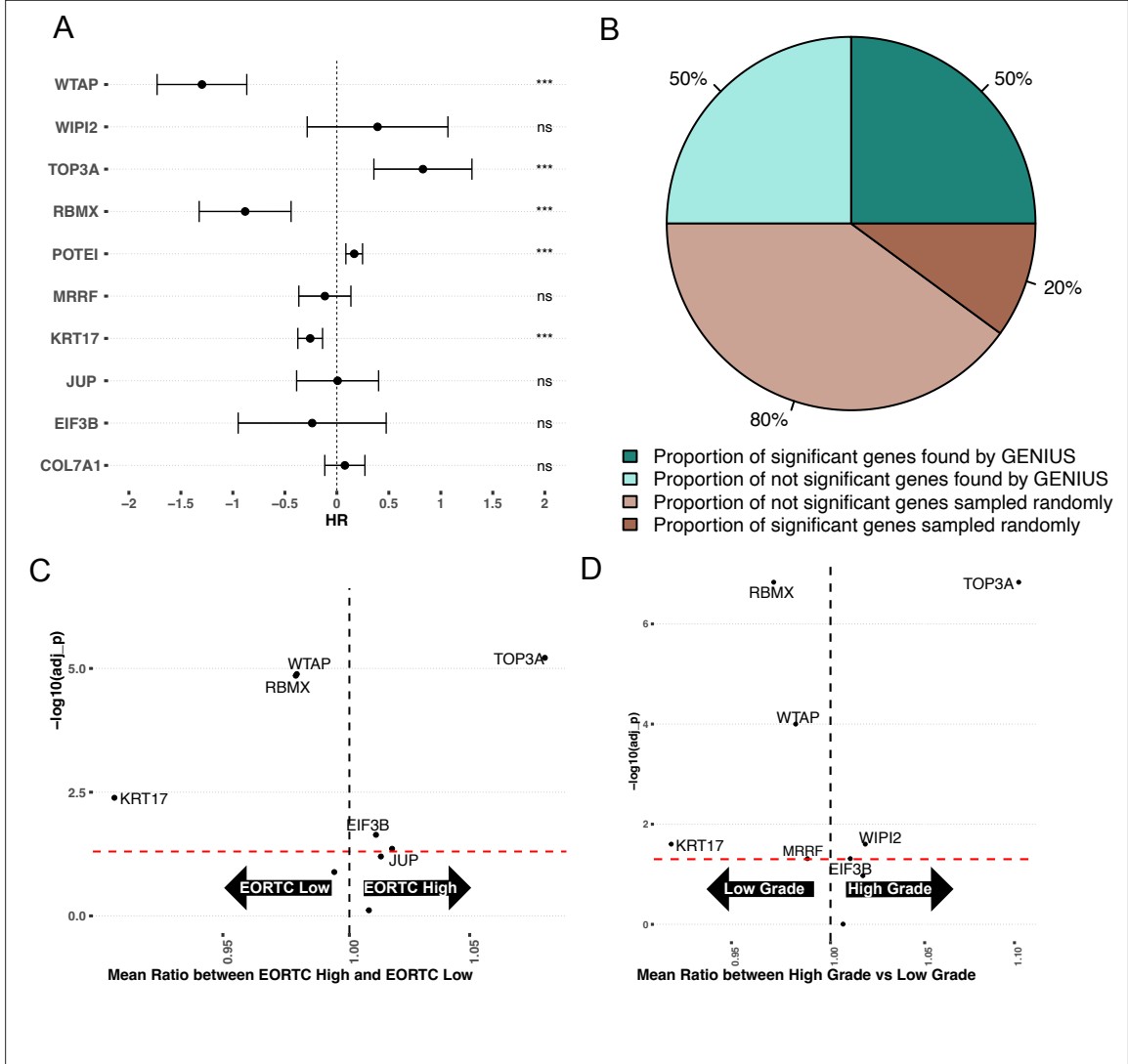

**Figure 5.** Validation on early-stage bladder cancer (UROMOL). (**A**) Forest plot showing top 10 expressed/methylated genes picked by GENIUS for BLCA. X-axis indicates Hazard Rate (HR). Stars indicates significance (* P < 0.05, ** P < 0.01, *** P < 0.001), "ns" indicates not significant (**B**) Comparison of median percent of randomly selected genes versus genes picked by GENIUS in cox proportional hazard model. (**C**) Volcano plot showing association of the top 10 expressed/methylated genes relative to EORTC-Low and EORTC-High groups. (**D**) Volcano plot showing association of the top 10 expressed/methylated genes relative to low- and high-grade BLCA tumors.

The online version of this article includes the following figure supplement(s) for figure 5:

**Figure supplement 1.** Histogram showing distribution of number of randomly picked significant genes in UROMOL dataset.

chance, only 20% of genes overall were found to be significantly associated with PFS compared to 50% of GENIUS genes (*Figure 5B*). To further investigate the top 10 genes picked by GENIUS, we compared the mean expression of each gene between different clinical risk groups (*EORTC, 2017*) and tumor grade. In this analysis, six of the 10 genes were significantly associated with EORTC status (*Figure 5C*, *Supplementary file 4*), and seven with grade (*Figure 5D*, *Supplementary file 5*).

## Discussion

In this work, we explored multiple options on how to transform multi-omics data into an image, leading to the utilization of deep learning models, which are often described as 'black box' models. The model architecture was evaluated in six different training scenarios, with a focus on validating the prediction of metastatic cancer. In this process, we also evaluated four different image layouts,

**Table 1.** Summary of BLCA genes in two validation cohorts.

| Gene | Early stage | Late stage (immunotherapy) | Description |
|---|---|---|---|
| TOP3A | HR > 0 (PFS), high grade, high EORTC | HR = 0 (OS), enriched in stage IV, enriched in CR/PR | Catalyses the transient breaking and rejoining of a single strand of DNA, involved in regulation of recombination and homology-directed repair. Positive association to OS in OV (*de Nonneville et al., 2022*) |
| RBMX | HR < 0 (PFS), low grade, low EORTC | HR > 0 (OS), enriched in CR/PR | Associated with translational control and DNA damage pathways. Reported to be negatively correlated with tumor stage, histological grade, and poor patient prognosis in BLCA (*Song et al., 2020*) |
| POTEI | HR = 0 (PFS) | HR = 0 (OS) | POTE family of proteins is associated with apoptotic cells (*Yu et al., 2023*) |
| KRT17 | HR < 0 (PFS), low grade, low EORTC | HR > 0 (OS), enriched in stages I–III, enriched in SD/PD | Associated with structural molecule activity and MHC class II receptor activity. Associated with metastasis and angiogenesis in variety of tumor types (*Ji et al., 2021*) |
| WIPI2 | HR = 0 (PFS), high grade | HR > 0 (OS), enriched in CR/PR | Component of the autophagy machinery that controls the major intracellular degradation process. WIPI2 is suggested as a biomarker for predicting colorectal cancer prognosis (*Yu et al., 2023*) |
| MRRF | HR = 0 (PFS), low grade | HR > 0 (OS) | Associated with the ribosome recycling factor, which is a component of the mitochondrial translational machinery. High expression is associated with poor outcome in ovarian cancer (*Song et al., 2020*) |
| EIF3B | HR = 0 (PFS), high EORTC | HR > 0 (OS) | Eukaryotic translation initiation factor 3 subunit B is a promoter associated with pancreatic cancer (*de Nonneville et al., 2022*) |
| JUP | HR = 0 (PFS) | HR > 0 (OS), enriched in stages I–III | Common junctional plaque protein. Controversial role in different malignancies. Knockdown of JUP in epithelium-like GC cells causes EMT and promotes GC cell migration and invasion (*Chen et al., 2021*) |
| WTAP | HR <0 (PFS), low EORTC, low grade | HR = 0 (OS) | Wilms' tumor 1-associating protein is associated to RNA methylation modifications, which regulate biological processes such as RNA splicing, cell proliferation, cell cycle, and embryonic development (*Chen et al., 2021*) |
| COL7A1 | HR = 0 (PFS) | HR > 0 (OS), enriched in SD/PD | Associated with metabolism of proteins and integrins in angiogenesis. Aberrant gene expression is associated with distinct tumor environment, metastasis and survival in multiple cancer types (*Oh et al., 2021*) |

concluding that of these, projecting the genome into a 198 × 198 square image with genes organized based on chromosome interaction (*Sarnataro et al., 2017*) performed the best. While that spatial organization improved the prediction, we recognize that there may exist a more optimal representation of multi-omics data which should be explored further in future work. Potential methods for organizing gene orientation in a multi-channel image could consider integrating topologically associating domains (*Beagan and Phillips-Cremins, 2020*) along with the spatial information from Hi-C. With the current implementation of GENIUS, gene layout can be set manually by the user to explore this issue further. For GENIUS, we have also included an auto-encoder in the network to recreate the input information without reconstruction loss. In this manner, the model itself can reconstruct the image of a genome in a format that is optimal for the prediction it is trying to make. The model also produces a latent representation of multi-omics data in a shape of a vector of a size 128 (L), which is later concatenated in a model when making final predictions. In order to investigate training effectiveness, we performed a UMAP clustering analysis of the L vector, where we compared the 2D representation of L with the variables of interest (*Figure 2C*). It is clear from this analysis that the L vector itself holds information that may be particularly relevant for multi-class prediction, but further analysis is needed to decipher what information is encoded in the L vector.

The main purpose behind the study was to demonstrate the feasibility of leveraging the power of deep learning techniques optimized for image analysis to interpret genome-derived multi-omics data. A key element of this approach includes the transformation of genomic data into images with genes arranged as pixels organized by chromosomal location. Beyond the readout from multi-omics data, this approach provides spatial information to the deep learning framework, which significantly improves the performance of the models (*Figure 2B*). To the best of our knowledge, we are the first to demonstrate the utility of spatial information and to provide a ready-to-use framework that incorporates spatial information and deep learning for the analysis of genome-derived multi-omics data. Furthermore, within the GENIUS framework, we facilitate the interpretation of the trained model

in order to explore the biology behind the prediction without the need for data preprocessing and multiple hypothesis correction. This was achieved by combining a deep learning network with IG (*Sundararajan et al., 2017*), allowing us to infer the attribution score for the input, resulting in non-parametric, ready-to-analyze output.

For every cancer type included in the dataset, we listed the top 10 genes driving metastatic disease and investigated in detail genes associated with BLCA metastasis and aggressiveness (*Table 1*). For this, we used two independent cohorts, one representing late-stage and metastatic cancer, and one representing early-stage cancer. In both cohorts, we tested if methylation and expression of genes found by the GENIUS framework were associated with survival at higher rates than when compared to randomly picked genes. In the late-stage BLCA cohort, seven out of 10 genes were significantly associated with overall survival, while in the early-stage BLCA cohort, we found that five out of 10 were significantly associated with PFS. That the results in the early-stage bladder cancer cohort (UROMOL) are less significant may relate to the model being trained to predict metastatic cancer. It is likely that the drivers of malignancy are different in early relative to late-stage disease, thus the top 10 genes found by GENIUS might not be prognostic in early-stage setting. In this regard, it is also worth noting that two of the top 10 genes (RBMX and KRT17) were associated with poor outcome in late-stage disease, while they were associated with improved outcome in early-stage disease. Interestingly, in the late-stage bladder cancer cohort, we observed that high expression of TOP3A associated with stage IV disease (*Figure 4C*). However, we also observed that high expression associated with improved response to immunotherapy. It is known that TOP3A has an important role in homology-directed repair and loss may be associated with chromosomal instability, which has shown a positive association with immunotherapy response (*Bakhoum and Cantley, 2018*; *Chen et al., 2022*; *Sokač et al., 2022*), potentially offering a likely explanation for this finding. Similarly, we observed that KRT17 was enriched in stages I–III, suggesting it may be associated with a less aggressive disease type. However, in the immunotherapy-treated cohort, KRT17 is associated with poor response to immunotherapy. In previous studies, KRT17 has been reported as associated with the development of metastatic disease, MHC type II receptor activity and angiogenesis (*Zhang et al., 2022*; *Ji et al., 2021*). This indicates that the KRT17 gene plays an important role as tumor suppressor gene in early-stage cancer, and that loss may further promote the development of aggressive, metastatic disease. While further research in this field is required to properly assess the utility of the reported genes, this work provides a framework that unlocks powerful machine-learning for more direct analysis of multi-omics data.

Taken together, we provide here the GENIUS framework along with analysis demonstrating the utility in multi-omics analysis. While we have focused on cancer analysis here, we believe GENIUS may find utility in a diverse range of genome-based multi-omics analyses. We have provided a github repository that can be used to transform data into images and train the same model predicting variables of user's interest and inferring the importance of input with respect to the desired output.

## Acknowledgements

NJB is a fellow of the Lundbeck Foundation (R272-2017-4040), and acknowledges funding from Aarhus University Research Foundation (AUFF-E-2018-7-14), and the Novo Nordisk Foundation (NNF21OC0071483). The results published here are in whole or part based upon data generated by the TCGA Research Network: https://www.cancer.gov/tcga.

## Additional information

### Funding

| Funder | Grant reference number | Author |
|---|---|---|
| Lundbeck Foundation | R272-2017-4040 | Nicolai J Birkbak |
| Aarhus Universitets Forskningsfond | AUFF-E-2018-7-14 | Nicolai J Birkbak |
| Novo Nordisk Fonden | NNF21OC0071483 | Nicolai J Birkbak |

| Funder | Grant reference number | Author |
|--------|------------------------|--------|

The funders had no role in study design, data collection, and interpretation, or the decision to submit the work for publication.

## Author contributions

Mateo Sokač, Conceptualization, Software, Formal analysis, Validation, Visualization, Writing – original draft, Writing – review and editing; Asbjørn Kjær, Validation, Writing – original draft; Lars Dyrskjøt, Supervision; Benjamin Haibe-Kains, Conceptualization, Supervision; Hugo JWL Aerts, Conceptualization, Writing – original draft; Nicolai J Birkbak, Conceptualization, Supervision, Funding acquisition, Writing – original draft, Project administration, Writing – review and editing

## Author ORCIDs

Mateo Sokač ⓘ http://orcid.org/0000-0001-9896-1544
Asbjørn Kjær ⓘ http://orcid.org/0009-0006-3307-0031
Nicolai J Birkbak ⓘ http://orcid.org/0000-0003-1613-9587

Reviewer #1 (Public Review): https://doi.org/10.7554/eLife.87133.3.sa1
Reviewer #2 (Public Review): https://doi.org/10.7554/eLife.87133.3.sa2
Author Response https://doi.org/10.7554/eLife.87133.3.sa3

# Additional files

## Supplementary files

• Supplementary file 1. Table shows the top 50 genes associated to metastatic disease development for every cancer type.

• Supplementary file 2. Summarized results for all output variables, containing top 10 events with respect to the predicted variable.

• Supplementary file 3. Top 10 expressed or methylated genes for every cancer type associated to metastatic disease.

• Supplementary file 4. Full model specifications for every multivariate cox proportional hazard model in late-stage immunotherapy-treated bladder cancer validation cohort (Mariathasan).

• Supplementary file 5. Full model specifications for every multivariate cox proportional hazard model in early-stage bladder cancer validation cohort (UROMOL).

• MDAR checklist

## Data availability

The data used for training the model in this analysis is publicly available through the Cancer Genome Atlas data portal. The validation cohorts (UROMOL and Mariathasan) are available under accession codes EGAS00001004693 and EGAS00001002556 from the European Genome-Phenome Archive.

The following previously published datasets were used:

| Author(s) | Year | Dataset title | Dataset URL | Database and Identifier |
|-----------|------|---------------|-------------|-------------------------|
| Sander C, Joshua M, Ozenberger BA, Ellrott K, Shmulevich I, Weinstein JN, Collisson EA, Mills GB, Shaw KM | 2013 | The Cancer Genome Atlas (TCGA) | https://www.ncbi.nlm.nih.gov/projects/gap/cgi-bin/study.cgi?study_id=phs000178.v11.p8 | dbGaP, phs000178.v11.p8 |

*Continued on next page*

*Continued*

| Author(s) | Year | Dataset title | Dataset URL | Database and Identifier |
|---|---|---|---|---|
| Lindskrog SV, Prip F, Taber A, Groeneveld CS, Birkenkamp-Demtröder K, Jensen JB, Strandgaard T, Nordentoft I, Christensen E, Sokac M, Birkbak NJ, Maretty L, Hermann GG, Petersen AC, Weyerer VG, rimm MO, Horstmann MS, jödahl GHöglund M, Steiniche T, Mogensen K, de Reyniès A, Nawroth R, Jordan B, Lin XD, ragicevic D, Ward DG, Goel A, Hurst CD, Raman JD, Warrick JI, Segersten U, Maurer T, Meeks JJ, DeGraff DJ, Bryan RT, Knowles MA, Simic TH, artmann AZ, warthoff EC, Malmström PU, Malats N, Real FX, Dyrskjøt L | 2021 | UROMOL 2020 | https://ega-archive.org/studies/EGAS00001004693 | EGA, EGAS00001004693 |
| Mariathasan S, Turley SJ, Nickles D, Castiglioni A, Yuen K, Wang Y, Kadel EE, Koeppen H, Astarita JL, Cubas R, Jhunjhunwala S, Banchereau R, Yang Y, Guan Y, Chalouni C, Ziai J, Şenbabaoğlu Y, Santoro S, Sheinson D, Hung J, Giltnane JM, Pierce AA, Mesh K, Lianoglou S, Riegler J, Carano RAD, Eriksson P, Höglund M, Somarriba L, Halligan DL, van der Heijden MS, Loriot Y, Rosenberg JE, Fong L, Mellman I, Chen DS, Green M, Derleth C, Fine GD, Hegde PS, Bourgon R, Powles T | 2018 | Mariathasan | https://ega-archive.org/studies/EGAS00001002556 | EGA, EGAS00001002556 |

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
