## [Editor Report · eLife assessment]

This **valuable** manuscript presents a new approach to transform multi-omics datasets into images and to exploit Deep Learning methods for image analysis of the transformed datasets. As an example, the method is applied to multi-omics datasets on different cancers. While the evidence in this specific case is **solid**, whether the method is working as advertised in other settings is not yet known.

---

## [Referee Report · Reviewer #1 (Public Review)]

This study by Sokač et al. entitled "GENIUS: GEnome traNsformatIon and spatial representation of mUltiomicS data" presents an integrative multi-omics approach which maps several genomic data sources onto an image structure on which established deep-learning methods are trained with the purpose of classifying samples by their metastatic disease progression signatures. Using published samples from the Cancer Genome Atlas the authors characterize the classification performance of their method which only seems to yield results when mapped onto one out of four tested image-layouts.

A few remaining issues are unclear to me:

1. While the authors have now extended the documentation of the analysis script they refer to as GENIUS, I assume that the following files are not part of the script anymore, since they still contain hard-coded file paths or hard-coded gene counts:

https://github.com/mxs3203/GENIUS/blob/master/GenomeImage/make_images_by_chr.py

https://github.com/mxs3203/GENIUS/blob/master/GenomeImage/randomize_normal_imgs.py

https://github.com/mxs3203/GENIUS/blob/master/GenomeImage/utils.py

If these files are indeed not part of the script anymore, then I would recommend removing them from the GitHub repo to avoid confusion. If, however, they are still part of the script, the authors failed to remove all hard-coded file paths and the software will fail when users attempt to use their own datasets.

2. The authors leave most of the data formatting to the user when attempting to use datasets other than their own presented for this study:

--clinical_data: Path to CSV file that must contain ID and label column we will use for prediction--ascat_data: Path to output matrix of ASCAT tool. Check the example input for required columns--all_genes_included: Path to the CSV file that contains the order of the genes which will be used to create Genome Image--mutation_data: Path CSV file representing mutation data. This file should contain Polyphen2 score and HugoSymbol--gene_exp_data: Path to the csv file representing gene expression data where columns=sample_ids and there should be a column named "gene" representing the HugoSymbol of the gene--gene_methyl_data: Path to the csv file representing gene methylation data wherecolumns=sample_ids and there should be a column named "gene1" representing the HugoSymbol of the gene

While this suggests that users will have a difficult time adjusting this analysis script to their own data, this issue is exacerbated by the fact that their analysis script has almost no internal checks whether data format standards were met. Thus, the user will be left with cryptic error messages and will likely give up soon after. I therefore strongly recommend adding internal data format checks and helpful error or warning messages to their script to guide users in the input data adoption process.

---

## [Referee Report · Reviewer #2 (Public Review)]

In this manuscript, Birkbak and colleagues use a novel approach to transform multi-omics datasets in images and apply Deep Learning methods for image analysis. Interestingly they find that the spatial representation of genes on chromosomes and the order of chromosomes based on 3D contacts leads to best performance. This supports that both 1D proximity and 3D proximity could be important for predicting different phenotypes. I appreciate that the code is made available as a github repository. The authors use their method to investigate different cancers and identify novel genes potentially involved in these cancers. Overall, I found this study important for the field.

In the original submission there were several major points with this manuscript could be grouped in three parts:

1. While the authors have provided validation for their model, it is not always clear that best approaches have been used. This has now been addressed in the revised version of the manuscript.

2. Potential improvement to the method

It is very encouraging the use of HiC data, but the authors used a very coarse approach to integrate it (by computing the chromosome order based on interaction score). We know that genes that are located far away on the same chromosome can interact more in 3D space than genes that are relatively close in 1D space. Did the authors consider this aspect? Why not group genes based on them being located in the same TAD? In the revised version of the manuscript, the authors discussed this possibility but did not do any new additional analysis.Authors claim that "given that methylation negatively correlates with gene expression, these were considered together". This is clearly not always the case. See for example https://genomebiology.biomedcentral.com/articles/10.1186/s13059-022-02728-5. In the revised version of the manuscript, the authors addressed fully this comment.

3. Interesting results that were not explained.

In Figure 3A methylation seems to be most important omics data, but in 3B, mutations and expression are dominating. The authors need to explain why this is the case. In the revised version of the manuscript, the authors have clarified this.

---

## [Author Response]

The following is the authors’ response to the original reviews.

**Reviewer #1 (Public Review):**
This study by Sokač et al. entitled "GENIUS: GEnome traNsformatIon and spatial representation of mUltiomicS data" presents an integrative multi-omics approach which maps several genomic data sources onto an image structure on which established deep-learning methods are trained with the purpose of classifying samples by their metastatic disease progression signatures. Using published samples from the Cancer Genome Atlas the authors characterize the classification performance of their method which only seems to yield results when mapped onto one out of four tested image-layouts.Major recommendations:In its current form, GENIUS analysis is neither computationally reproducible nor are the presented scripts on GitHub generic enough for varied applications with other data. The GENIUS GitHub repository provides a collection of analysis scripts and not a finished software solution (e.g. command line tool or other user interface) (the presented scripts do not even suffice for a software prototype). In detail, the README on their GitHub repository is largely incomplete and reads analogous to an incomplete and poorly documented analysis script and is far from serving as a manual for a generic software solution (this claim was made in the manuscript).

We apologize for this oversight, and we have now invested considerable resources into making the documentation more detailed and accurate. We have created a new GitHub repository (https://github.com/mxs3203/GENIUS) that contains a small set of example data and all the necessary scripts to run GENIUS. The README file guides the user through each step of the GENIUS framework but it also contains a bash script that runs all the steps at once. When a user would like to use it on their own data, they need to replace the input data with their data but in the same format as the example input data. This is now fully documented in the README file. All scripts have arguments that can be used to point to custom data. The entire pipeline using example data can be run using run_genius.sh script. This script will produce CSV files and PNG files inside the ExtractWithIG folder containing attribution scores for every cancer type tested.

The authors should invest substantially into adding more details on how data can be retrieved (with example code) from the cited databases and how such data should then be curated alongside the input genome to generically create the "genomic image".

Data for analysis can be sourced from multiple locations, what we have used in our examples and for development was based on data from the TCGA. It can be retrieved from the official TCGA data hub or through Xena Browser (https://xenabrowser.net/). However, the data formats are generic, and similar data types (mutation, expression, methylation, copy number) can be obtained from multiple sources. We have added example data to demonstrate the layout, and we have a script included that creates the layout from standard mutation, expression, methylation and copy number data formats. We have substantially improved the annotations, including detailed descriptions of the data layout along with examples, and we have, as part of our validation, had an independent person test run the scripts using TCGA example data we provided on the new GitHub page.

In addition, when looking at the source code, parameter configurations for training and running various modules of GENIUS were hard-coded into the source code and users would have to manually change them in the source code rather than as command line flags in the software call. Furthermore, file paths to the local machine of the author are hard-coded in the source code, suggesting that images are sourced from a local folder and won't work when other users wish to replicate the analysis with other data. I would strongly recommend building a comprehensive command line tool where parameter and threshold configurations can be generically altered by the user via command line flags.

Apologies, we have changed the code and removed all hard-coded paths. All paths are now relative to the script using them. Furthermore, we made the config file more visible and easier to use. The example run can be found on the new github repository we linked in the previous comment.

We also inserted the following text in the manuscript

The GitHub repository contains example data and instructions on how to use the GENIUS framework.

A comprehensive manual would need to be provided to ensure that users can easily run GENIUS with other types of input data (since this is the claim of the manuscript). Overall, due to the lack of documentation and hard-coded local-machine folder paths it was impossible to computationally reproduce this study or run GENIUS in general.

Apologies, we have completely reworked the code base, and extensively annotated the code. We have also made highly detailed step-by-step instructions that should enable any user to run GENIUS on their own or public data.

In the Introduction the authors write: "To correct for such multiple hypothesis testing, drastic adjustments of p-values are often applied which ultimately leads to the rejection of all but the most significant results, likely eliminating a large number of weaker but true associations.". While this is surely true for any method attempting to separate noise from signal, their argument fails to substantiate how their data transformation will solve this issue. Data transformation and projection onto an image for deep-learning processing will only shift the noise-to-signal evaluation process to the postprocessing steps and won't "magically" solve it during training.

The data transformation does not solve the problem of multiple hypothesis testing but it facilitates the use of computer vision algorithms and frameworks on rich multi-omics data. Importantly, transforming the data into genome images, training the model, and inspecting it with integrated gradients can be interpreted as running a single test on all of the data.

Analyzing multiomics data using classical statistical methods typically means that we perform extensive filtering of the data, removing genes with poor expression/methylation/mutation scores, and then e.g. perform logistic regression against a desired outcome, or alternatively, perform multiple statistical tests comparing each genomic feature independently against a desired outcome. Either way, information is lost during initial filtering and we must correct the analysis for each statistical test performed. While this increases confidence in whichever observation remains significant, it also undoubtedly means that we discard true positives. Additionally, classical statistical methods such as those mentioned here do not assume a spatial connection between data points, thus any relevant information relating to spatial organization is lost.

Instead, we propose the use of the GENIUS framework for multiomics analysis. The GENIUS framework is based on deep neural nets and relies on Convolutions and their ability to extract interactions between the data points. This particularly considers spatial information, which is not possible using classical statistical methods such as logistic regression where the most similar approach to this would include creating many models with many interactions.

Furthermore, integrated gradients is a non-parametric approach that simply evaluates the trained model relative to input data and output label, resulting in attribution for each input with respect to the output label. In other words, integrated gradients represent the integral of gradients with respect to inputs along the path from a given baseline to input. The integral is described in Author response image 1:

More about integrated gradients can be read on the Captum webpage(https://captum.ai/docs/introduction) or in original paper https://arxiv.org/abs/1703.01365.

Since we transformed the data into a data structure (genome image) that assumes a spatial connection between genes, trained the model using convolutional neural networks and analyzed the model using integrated gradients, we can treat the results without any parametric assumption. As a particular novelty, we can sort the list based on attribution score and take top N genes as our candidate biomarkers for the variable of interest and proceed with downstream analysis or potentially functional validation in an in vitro setting. In this manner, the reviewer is correct that the signal-to-noise evaluation is shifted to the post-processing steps. However, the benefit of the GENIUS framework is particularly that it enables integration of multiple data sources without any filtering, and with constructing a novel data structure that facilitates investigation of spatial dependency between data points, thus potentially revealing novel genes or biomarkers that were previously removed through filtering steps. However, further downstream validation of these hits remains critical.

We added the following paragraph to make this more clear

"Integrated Gradients is a non-parametric approach that evaluates the trained model relative to input data and output label, resulting in attribution scores for each input with respect to the output label. In other words, Integrated Gradients represent the integral of gradients with respect to inputs along the path from a given baseline. By using Integrated Gradients, we provide an alternative solution to the problem posed by performing multiple independent statistical tests. Here, instead of performing multiple tests, a single analysis is performed by transforming multiomics data into genome images, training a model, and inspecting it with Integrated Gradients. Integrated Gradients will output an attribution score for every gene included in the genome image and those can be ranked in order to retrieve a subset of the most associated genes relative to the output variable."

In addition, multiple-testing correction is usually done based on one particular data source (e.g.expression data), while their approach claims to integrate five very different genomic data sources with different levels and structures of technical noise. How are these applications comparable and how is the training procedure able to account for these different structures of technical noise? Please provide sufficient evidence for making this claim (especially in the postprocessing steps after classification).

The reviewer is correct that there will be different technical noise for each data source. However, each data source is already processed by standardized pipelines used for interpreting sequence-level data into gene expression, mutations, copy number alterations and methylation levels. Thus, sequence-level technical noise is not evaluated as part of the GENIUS analysis. Nevertheless, the reviewer is correct that sample-level technical noise, such as low tumor purity or poor quality sequencing, undoubtedly can affect the GENIUS predictions, as is true for all types of sequence analysis. As part of GENIUS, an initial data preprocessing step (which is performed automatically as part of the image generation), is that each data source is normalized within that source and linearly scaled in range zero to one (min-max scaling). This normalization step means that the impact of different events within and between data sources are comparable since the largest/smallest value from one data source will be comparable to the largest/smallest value from another data source.

Additionally, deep neural networks, particularly convolutional networks, have been shown to be very robust to different levels of technical noise (Jang, McCormack, and Tong 2021; Du et al. 2022). In the manuscript we show the attribution scores for different cancer types in figure 3B of the paper. Here, the top genes include established cancer genes such as P53, VHL, PTEN, APC and PIK3CA, indicating that the attribution scores based on GENIUS analysis is a valid tool to identify potential genes of interest. Furthermore, when focusing the analysis on predicting metastatic bladder cancer, we were able to show that of the top 10 genes with the highest attribution scores, 7 showed significant association with poor outcome in an independent validation cohort of mostly metastatic patients (shown in figure 4).

I didn't find any computational benchmark of GENIUS. What are the computational run times, hardware requirements (e.g. memory usage) etc that a user will have to deal with when running an analogous experiment, but with different input data sources? What kind of hardware is required GPUs/CPUs/Cluster?

We apologize for not including this information in the manuscript. We added the following section in to the manuscript:

"Computational Requirements

In order to train the model, we used the following hardware configuration: Nvidia RTX3090 GPU, AMD Ryzen 9 5950X 16 core CPU, and 32Gb of RAM memory. In our study, we used a batch size of 256, which occupied around 60% of GPU memory. Training of the model was dependent on the output variable. For metastatic disease prediction, we trained the model for approximately 4 hours. This could be changed since we used early stopping in order to prevent overfitting. By reducing the batch size to smaller numbers, the technical requirements are reduced making it possible to run GENIUS on most modern laptops."

A general comment about the Methods section: Models, training, and validation are very vaguely described and the source code on GitHub is very poorly documented so that parameter choices, model validation, test and validation frameworks and parameter choices are neither clear nor reproducible.

Apologies, we have updated the methods section with more details on models, training and validation. Additionally, we have moved the section on evaluating model performance from the methods section to the results section, with more details on how training was performed.

We also agree that the GitHub page is not sufficiently detailed and well structured. To remedy this, we have made a new GitHub page that only has the code needed for analysis, example input data, example runs, and environment file with all library versions. The GitHub repository is also updated in the manuscript.

The new GitHub page can be found on: https://github.com/mxs3203/GENIUS

Please provide a sufficient mathematical definition of the models, thresholds, training and testing frameworks.

We sincerely apologize, but we do not entirely follow the reviewers request on this regard. The mathematical definitions of deep neural networks are extensive and not commonly included in research publications utilizing deep learning. We have used PyTorch to implement the deep neural net, a commonly used platform, which is now referenced in the methods. The design of the deep learning network used for GENIUS is described in figure 1, and the relevant parameters are described in methods. The hyper parameters are described in the methods section, and are as follows:

"All models were trained with Adagrad optimizer with the following hyperparameters: starting learning rate = 9.9e-05 (including learning rate scheduler and early stopping), learning rate decay and weight decay = 1e-6, batch size = 256, except for memory-intensive chromosome images where the batch size of 240 was used."

In chapter "Latent representation of genome" the authors write: "After successful model training, we extracted the latent representations of each genome and performed the Uniform Manifold Approximation and Projection (UMAP) of the data. The UMAP projected latent representations into two dimensions which could then be visualized. In order to avoid modeling noise, this step was used to address model accuracy and inspect if the model is distinguishing between variables of interest.". In the recent light of criticism when using the first two dimensions of UMAP projections with omics data, what is the evidence in support of the author's claim that model accuracy can be quantified with such a 2D UMAP projection? How is 'model accuracy' objectively quantified in this visual projection?

We apologize for not clarifying this. The UMAP was done on L, the latent vector, which by assumption should capture the most important information from the “genome image”. In order to confirm this, we plotted the first two dimensions of UMAP transformation and colored the points by the output variable. If the model was capturing noise, there should not be any patterns on the plot (randomized cancer-type panel). Since, in most cases, we do see an association between the first two UMAP dimensions and the output variable, we were confident that the model was not modeling (extracting) noise.

To clarify this, we changed the sentence in the manuscript so it is more clear that this is not an estimation of accuracy but only an initial inspection of the models:

The UMAP projected latent representations into two dimensions which could then be visualized. In order to avoid modeling noise, this step was used to inspect if the model is distinguishing between variables of interest.

In the same paragraph "Latent representation of genome" the authors write: "We observed that all training scenarios successfully utilized genome images to make predictions with the exception of Age and randomized cancer type (negative control), where the model performed poorly (Figure 2B).". Did I understand correctly that all negative controls performed poorly? How can the authors make any claims if the controls fail? In general, I was missing sufficient controls for any of their claims, but openly stating that even the most rudimentary controls fail to deliver sufficient signals raises substantial issues with their approach. A clarification would substantially improve this chapter combined with further controls.

We apologize for not stating this more clearly. Randomized cancer type was used as a negative control since we expect that model would not be able to make sense of the data if predicting randomized cancer type. As expected, the model failed to predict the randomized cancer types. This can be seen in Figure 2C, where UMAP representations (based on the latent representation of the data, the vector L) are made for each output variable. Not seeing any patterns in UMAP shows that, as expected, the model does not know how to extract useful information from “genome image” when predicting randomized cancer type (as when randomly shuffling the labels there is no genomic information to decipher). Similar patterns were observed for Age, indicating that patient age cannot be determined from the multi-omics data. Conversely, when GENIUS was trained against wGII, TP53, metastatic status, and cancer type, we observed that samples clustered according to the output label.

**Reviewer #2 (Public Review):**
In this manuscript, Birkbak and colleagues use a novel approach to transform multi-omics datasets in images and apply Deep Learning methods for image analysis. Interestingly they find that the spatial representation of genes on chromosomes and the order of chromosomes based on 3D contacts leads to best performance. This supports that both 1D proximity and 3D proximity could be important for predicting different phenotypes. I appreciate that the code is made available as a github repository. The authors use their method to investigate different cancers and identify novel genes potentially involved in these cancers. Overall, I found this study important for the field.The major points of this manuscript could be grouped in three parts:1. While the authors have provided validation for their model, it is not always clear that best approaches have been used.a) In the methods there is no mention of a validation dataset. I would like to see the authors training on a cancer from one cohort and predict on the same cancer from a different cohort. This will convince the reader that their model can generalise. They do something along those lines for the bladder cancer, but no performance is reported. At the very least they should withhold a percentage of the data for validation. Maybe train on 100 and validate on the remaining 300 samples. They might have already done something along these lines, but it was not clear from the methods.

Apologize for not being sufficiently clear in the manuscript. We did indeed validate the performance within the TCGA cohort, using holdout cross validation. Here, we trained the network on 75% of the cohort samples (N = 3825), and tested on the remaining 25% (N = 1276).

To make this more clear, we have rewritten section “GENIUS classification identifies tumors likely to become metastatic” as such:

"The omics data types included somatic mutations, gene expression, methylation, copy number gain and copy number loss. Using holdout type cross-validation, where we split the data into training (75%) and validation (25%), we observed a generally high performance of GENIUS, with a validation AUC of 0.83 for predicting metastatic disease (Figure 2B)."

We also added the following sentence in the legend of Figure 2:

"The x-axis represents epochs and y-axis represents AUC score of fixed 25% data we used for accuracy assessment within TCGA cohort."

The accuracy of GENIUS could not be validated on the other two bladder cohorts since they do not contain all the data for the creation of five-dimensional genome images. However, we were able to investigate if the genes with the highest attribution scores towards metastatic bladder cancer obtained based on the TCGA samples also showed a significant association with poor outcome in the two independent bladder cancer cohorts. Here, we observed that of the top 10 genes with the highest attribution scores, 5 were associated with poor outcome in the early stage bladder cancer cohort, and 7 were associated with poor outcome in the late stage/metastatic bladder cancer cohort.

b) It was not clear how they used "randomised cancer types as the negative control". Why not use normal tissue data or matched controls?

In the study, we built six models, one for each variable of interest. One of them was cancer type which performed quite well. In order to assess the model on randomized data, we randomized the labels of cancer type and tried predicting that. This served as “negative control” since we expected the model to perform poorly in this scenario. To make this more clear in the manuscript, we have expanded the description in the main text. We have also added the description of this to each supplementary plot to clarify this further.

While normal tissue and matched controls would have been an optimal solution, unfortunately, such data is not available.

c) If Figure 2B, the authors claim they have used cross validation. Maybe I missed it, but what sort of cross validation did they use?

We apologize for not being sufficiently clear. As described above, we used holdout cross-validation to train and evaluate the model. We clarified this in the text:

"Using holdout type cross-validation, where we split the data into training (80%) and validation(20%), we observed a generally high performance of GENIUS, with a mean validation AUC of0.83 (Figure 2B)"

1. Potential improvement to the methoda) It is very encouraging the use of HiC data, but the authors used a very coarse approach to integrate it (by computing the chromosome order based on interaction score). We know that genes that are located far away on the same chromosome can interact more in 3D space than genes that are relatively close in 1D space. Did the authors consider this aspect? Why not group genes based on them being located in the same TAD?

We thank the reviewer for this suggestion and we will start looking into how to use TAD information to create another genome representation. In this study, we tried several genome transformations, which proved to be superior compared to a flat vector of features (no transformation). We are aware that squared genome transformation might not be optimal, so we designed the network that reconstructs the genome image during the training. This way, the genome image is optimized for the output variable of choice by the network itself. However, we note that the order of the genes themselves, while currently based on HiC, can be changed by the user. The order is determined by a simple input file which can be changed by the user with the argument “all_genes_included”. Thus, different orderings can be tested within the overall square layout. This is now detailed in the instructions on the new GitHub page.

The convolutional neural network uses a kernel size of 3x3, which captures the patterns of genes positioned close to each other but also genes that are far away from each other (potentially on another chromosome). Once convolutions extract patterns from the image, the captured features are used in a feed-forward neural network that makes a final prediction using all extracted features/patterns regardless of their location in the genome image.

We also inserted the following sentence in discussion:

"Given that spatial organization improved the prediction, we recognize that there may exist a more optimal representation of multi-omics data which should be explored further in future work. Potential methods for organizing gene orientation in a 2D image could consider integrating topologically associating domains[39] along with the spatial information from HiC. This is already possible to explore with the current implementation of GENIUS, where gene layout can be set manually by the user."

b) Authors claim that "given that methylation negatively correlates with gene expression, these were considered together". This is clearly not always the case. See for example https://genomebiology.biomedcentral.com/articles/10.1186/s13059-022-02728-5. What would happen if they were not considered together?

We thank the reviewer for this insightful comment. We agree with the reviewer that methylation does not always result in lower expression, although methylation levels in most cases should correlate negatively to RNA expression, but with a gene-specific factor. Indeed, there are tools developed that infer RNA expression based on methylation, making use of gene-specific correction factors. E.g. Mattesen et al (Mattesen, Andersen, and Bramsen 2021).

However, upon reflection we agree with the reviewer that we cannot assume for all genes that methylation equals low expression. Therefore, we have performed an analysis where we compared the methylation level to gene expression levels for all tested genes within bladder cancer. We computed Pearson’s correlation of 16,456 genes that have both methylation and expression scores. Of these, 8528 showed a negative correlation. After p-value correction, this resulted in 4774 genes where methylation was significantly negatively associated with expression. For these genes we performed the subsequent analysis in bladder cancer, where methylation and expression were considered together. This updated analysis has been included in supplementary figure 10, and the results section has been amended to reflect this. Overall, this analysis resulted in 4 of 10 genes being replaced in the downstream analysis. However, we note that the final results did not materially change, nor did the conclusions.

**Author response image 2. sa3fig2:** Correlation between gene-level methylation and gene expression in TCGA BLCA cohort.

1. Interesting results that were not explained.a) In Figure 3A methylation seems to be the most important omics data, but in 3B, mutations and expression are dominating. The authors need to explain why this is the case.

We apologize for not explaining this in more detail. Figure 3B shows the attribution scores scaled within the cancer type, where Figure 3A shows raw attribution scores for each data source included. The reason for this is that methylation and expression have in general, smaller attribution scores but more events where a single mutation often is characterized with large attribution scores and the rest of them with very small attribution. In order to make those numbers comparable and take into account biological differences between the cancer type, we scaled the scores within each cancer type.

To make this more clear we modified the first sentence in “Interpreting the GENIUS model classifying metastatic cancer biology” section:

"Analysing raw attribution scores we concluded the most informative data type overall regarding the development of metastatic disease was methylation (Figure 3A).…We also noticed that mutation data often had a single mutation with large attribution score where expression and methylation showed multiple genes with high attribution scores… … The normalization step is crucial to make results comparable as underlying biology is different in each cancer type included in the study."

**Reviewer #1 (Recommendations For The Authors):**
While I appreciate the creative acronym of the presented software solution (GENIUS), it may easily be confused with the prominent software Geneious | Bioinformatics Software for Sequence Data Analysis which is often employed in molecular life science research. I would suggest renaming the tool.

We appreciate the comment but prefer to keep the name. Given that the abbreviation is not exactly the same and the utility is different, we are confident that there will be no accidental mixup between these tools.

A huge red flag is the evaluation of the input image design which clearly shows that classification power after training is insufficient for three out of four image layouts (and even for the fourth AUC is between 0.70-0.84 depending on the pipeline step and application). Could the authors please clarify why this isn't cherry-picking (we use the one layout that gave some form of results)? In light of the poor transformation capacity of this multi-omics data onto images, why weren't other image layouts tried and their classification performance assessed? Why should a user assume that this image layout that worked for this particular input dataset will also work with other datasets if image transformation is performing poorly in most cases?

We apologize for not describing this further in the manuscript. We wrote in the manuscript that we could not know what genome representation is optimal as it is difficult to know. A flat vector represents a simple (or no) transformation since we simply take all of the genes from all of the data sources and append them into a single list. Chromosome image and square image are two transformations we tried, and we focused on the square image since in our hands it showed superior performance relative to other transformations.

**Reviewer #2 (Recommendations For The Authors):**
Minor points:1. Legends of supplementary Figures are missing.

We thank the reviewer for this comment and apologize for missing it. All legends have been added now.

1. For some tests the authors use F1 score while for other AUC, they should be consistent. Report all metrics for all comparisons or report one and justify why that only metric.

We apologize for not being sufficiently clear. AUC is a standard score used for binary classification, while the F1 score is used for multiclass classification. We have now described this in the methods section, and hope this is now sufficiently clear.

"When predicting continuous values, the model used the output from the activation function with the mean squared error loss function. When predicting multi-class labels, the performance measure was defined by the F1 score, a standard measure for multiclass classification that combines the sensitivity and specificity scores and is defined as the harmonic mean of its precision and recall. To evaluate model performance against the binary outcome, ROC analysis was performed, and the area under the curve (AUC) was used as the performance metric."

1. not sure how representation using UMAP in Figure 2C is helping understand the performance.

Apologies for the poor wording in the results section. The purpose of the UMAP representation was to visually inspect if the model was distinguishing between variables of interest, not to estimate model performance. We have rephrased the text in the methods section to make this clear:

"After successful model training, we extracted the latent representations of each genome and performed the Uniform Manifold Approximation and Projection (UMAP) of the data for the purpose of visual inspection of a model."

And

"In order to avoid modeling noise, this step was used to inspect if the model is distinguishing between variables of interest."

And also in the results section:

"In order to visually inspect patterns captured by the model, we extracted the latent representations of each genome and performed the Uniform Manifold Approximation and Projection (UMAP) of the data to project it into two dimensions."

1. Instead of pie chart in 3A, the authors should plot stacked barplots (to 100%) so it would be easier to compare between the different cancer types.

We thank the reviewer for the suggestion; however, since we wanted to compare the relative impact of each data source with each other, we used pie charts. Piecharts are often better for describing relative values, whereas bar plots are better for absolute values.

References

Du, Ruishan, Wenhao Liu, Xiaofei Fu, Lingdong Meng, and Zhigang Liu. 2022. “Random Noise Attenuation via Convolutional Neural Network in Seismic Datasets.” Alexandria Engineering Journal 61 (12): 9901–9.

Jang, Hojin, Devin McCormack, and Frank Tong. 2021. “Noise-Trained Deep Neural Networks Effectively Predict Human Vision and Its Neural Responses to Challenging Images.” PLoS Biology 19 (12): e3001418.

Mattesen, Trine B., Claus L. Andersen, and Jesper B. Bramsen. 2021. “MethCORR Infers GeneExpression from DNA Methylation and Allows Molecular Analysis of Ten Common Cancer Types Using Fresh-Frozen and Formalin-Fixed Paraffin-Embedded Tumor Samples.” Clinical Epigenetics 13 (1): 20.